# Structural and Dynamic Disturbances Revealed by Molecular Dynamics Simulations Predict the Impact on Function of CCT5 Chaperonin Mutations Associated with Rare Severe Distal Neuropathies

**DOI:** 10.3390/ijms24032018

**Published:** 2023-01-19

**Authors:** Federica Scalia, Giosuè Lo Bosco, Letizia Paladino, Alessandra Maria Vitale, Leila Noori, Everly Conway de Macario, Alberto J. L. Macario, Fabio Bucchieri, Francesco Cappello, Fabrizio Lo Celso

**Affiliations:** 1Department of Biomedicine, Neuroscience and Advanced Diagnostics (BiND), University of Palermo, 90127 Palermo, Italy; 2Euro-Mediterranean Institute of Science and Technology (IEMEST), 90139 Palermo, Italy; 3Department of Mathematics and Computer Science, University of Palermo, 90123 Palermo, Italy; 4Department of Anatomy, School of Medicine, Tehran University of Medical Science, Tehran 1417653911, Iran; 5Department of Microbiology and Immunology, School of Medicine, University of Maryland at Baltimore—Institute of Marine and Environmental Technology (IMET), Baltimore, MD 21202, USA; 6Department of Physics and Chemistry—Emilio Segrè, University of Palermo, 90128 Palermo, Italy; 7Ionic Liquids Laboratory, Institute of Structure of Matter, Italian National Research Council (ISM-CNR), 00133 Rome, Italy

**Keywords:** chaperone system, CCT5, CCT5 mutations, CCT5 chaperonopathies, apical domain, hydrogen bonds, electrostatic potential, protein binding

## Abstract

Mutations in genes encoding molecular chaperones, for instance the genes encoding the subunits of the chaperonin CCT (chaperonin containing TCP-1, also known as TRiC), are associated with rare neurodegenerative disorders. Using a classical molecular dynamics approach, we investigated the occurrence of conformational changes and differences in physicochemical properties of the CCT5 mutations His147Arg and Leu224Val associated with a sensory and a motor distal neuropathy, respectively. The apical domain of both variants was substantially but differently affected by the mutations, although these were in other domains. The distribution of hydrogen bonds and electrostatic potentials on the surface of the mutant subunits differed from the wild-type molecule. Structural and dynamic analyses, together with our previous experimental data, suggest that genetic mutations may cause different changes in the protein-binding capacity of CCT5 variants, presumably within both hetero- and/or homo-oligomeric complexes. Further investigations are necessary to elucidate the molecular pathogenic pathways of the two variants that produce the two distinct phenotypes. The data and clinical observations by us and others indicate that CCT chaperonopathies are more frequent than currently believed and should be investigated in patients with neuropathies.

## 1. Introduction

The chaperone system (CS) of an organism consists of all its molecular chaperones, co-chaperones, chaperone co-factors, and chaperone interactors and receptors [1]. The canonical functions of the CS are directed at maintaining protein homeostasis. Its partners include the ubiquitin-proteasome system (UPS) and the chaperone-mediated autophagy machinery. The CS also has non-canonical functions relating to inflammatory and autoimmune conditions and cancer and interacts with the immune system (IS) [2,3,4,5,6,7,8,9]. The chief components of the CS are the molecular chaperones, encompassing a large variety of molecules that can be classified according to their molecular weight. Some, referred to as Hsp’s (heat shock proteins), are members of phylogenetically related families [10]. One of these is the CCT (chaperonin-containing TCP-1) family of chaperonins, also called TRiC (TCP-1 ring complex), with molecular weights within the range of 55-64 kDa. This family is composed of 14 members, of which nine, named CCT subunits 1, 2, 3, 4, 5, 6A, 6B, 7, and 8, can assemble to form the CCT hexadecameric complex, with chaperone functions in humans and many other species [11]. Chaperones, including chaperonins, are typically cytoprotective, but, if abnormal in quantity/distribution/structure/function, they can become etiopathogenic and cause diseases, the chaperonopathies [12,13]. These diseases can be acquired or genetic, the latter caused by pathogenic variants of the chaperone genes. In our previous work, we discussed several rare genetic chaperonopathies related to disorders of the nervous system [14]. For example, mutations in the CCT5 subunit are associated with serious distal neuropathies [15,16]. Although the clinical and genetic aspects of many chaperonopathies have been described, the molecular mechanisms underpinning the lesions found in patients are still poorly understood [17]. Moreover, the impact of the pathogenic mutations on the structure/function of the affected chaperones has not been elucidated in most cases. Therefore, if progress is to be made in the diagnosis and management of genetic chaperonopathies, it is necessary to clarify the effects of the mutations on the structure/function of the chaperone molecules and to characterize the mechanisms causing the histological lesions associated with them. We are investigating these issues by studying the effects of two CCT5 subunit mutations, Leu224Val and His147Arg, associated with an early onset distal motor neuropathy [16] and an autosomal recessive mutilating sensory neuropathy with spastic paraplegia [15], respectively. We characterized the histopathological lesions of the striated muscle in a patient with the Leu224Val mutation and found various abnormalities, most likely caused by a defective chaperoning function of the CCT complex bearing the CCT5 mutant subunit [18].

Of note is the importance of the CCT5 subunits, not only in the constitution of the whole CCT complex, but also their existence and function as free monomers and as homo-oligomeric complexes [19,20,21]. Depletion of the CCT5 subunit in mammalian cells is related to their morphological changes [22], and CCT5 monomers are involved in actin folding and actin transcriptional control by modulating the myocardin-related co-transcription factor-A (MRTFA)/serum response factor (SRF) pathway [23]. CCT5 monomers can assemble into homo-oligomeric complexes, which is also true for CCT4 monomers. However, only the CCT5 homo-oligomeric complex can act as an initial step in the assembly of the full hetero-oligomeric complex by assembling with all the other subunits (following the order: CCT2, CCT4, CCT1, CCT3, CCT7, CCT8, CCT6). Therefore, the homo-oligomeric complex CCT5 has been proposed as a template or intermediate form for the assembly of the whole, functionally active, CCT complex [24].

Human CCT5 consists of 541 amino acids organized in β-sheets and α-helixes distributed into three domains, apical, equatorial, and intermediate (Appendix A). The apical domain, which comprises amino acids 227 to 378, includes α-helix-9 that functions as a built-in lid to close the folding chamber α-helix-10 that is the substrate-binding interface, along with the proximal ring base region [25]. The equatorial domain consists of the *N*-terminal (amino acids 1–154) and *C*-terminal (amino acids 418–541) segments that form loop-A and loop-B for the interaction with the α- and β-phosphate groups of ADP and a sensor loop, also called a stem-loop, located between the β1 and β2 sheets, for the contacts with the substrate (e.g., tubulin) (Appendix A) [25]. The β1 and β2 sheets also form inter-monomer interactions within the hexadecameric complex, contacting the equatorial β13 sheet of the adjacent subunit (Appendix A). The His147Arg mutation occurs in the α-helix 5 of the equatorial domain (Appendix A).

The Leu224Val mutation occurs in the β4-sheet of the intermediate domain (Appendix A). The intermediate domain consists of two parts, the *N*-terminal (amino acids 155–226) and the *C*-terminal (amino acids 381–417) segments, including the α-helixes of the lower hinge, helix-6 and helix-12, which establish contact with the adjacent segments of the equatorial domain, whereas helix-7 contacts the upper hinge. The upper hinge consists of β-sheets, β3, β4, β12 (Appendix A).

Earlier bioinformatics results predicted that the Leu224Val mutation, although positioned in the intermediate domain, would have a pronounced impact on the apical domain [15]. The His147Arg mutation was modelled on the archaeal homolog Cpn60, which has high sequence and structural homology to human CCT5 [26]. The main effects of the mutation on the “humanized” subunit were: (1) steric hindrances and helix–helix charge repulsions that could impair the conformational change needed for ATP binding; (2) decrease in the formation of the double ring hetero-oligomeric complex; (3) impairment of the chaperoning functions at suboptimal temperatures. A loss of flexibility of the mutated CCT5 subunit was associated with impairment of the ability to protect cells from the formation of amyloid fibrils [26]. The His147Arg mutation affects the biochemical properties of the CCT5 homo-oligomeric ring when expressed in *Escherichia coli*, causing a charge change in the CCT5 subunits and reducing the folding efficiency of the mutant CCT5 complex [21].

Here, we report the structural and dynamic disturbances revealed by molecular dynamics simulations associated with the Leu224Val and His147Arg mutants.

## 2. Results

### 2.1. Impact of the His147Arg and Leu224Val Mutations on the CCT5 Structure

We assessed the impact of His147Arg and Leu224Val mutations on the structure-conformation of the CCT5 subunit by molecular dynamics simulations, comparing the wild-type and the two mutants under three conditions: nucleotide-free, ATP-bound, and ADP-bound. The structures of the three molecules differed in the three conditions; the differences were more notable in the nucleotide-free and ATP-bound conditions (Figure 1A,B) than in the ADP-bound condition (Figure 1C) (see Appendix A). Although the mutations occur in the equatorial (His147Arg) or the intermediate (Leu224Val) domains, the apical domain was the most delocalized: relaxed in the Leu224Val variant and folded back on itself in the His147Arg mutant. The delocalization of the apical domains was maintained during the simulations as the proteins reached equilibrium.

### 2.2. Predicted Fluctuation of Amino Acids in the Apical and Intermediate Domains of the Two Mutants Compared with the Wild Type

The initial position, the functional fluctuation, and thus the stability and flexibility of the apical domain of the CCT5 monomer alone and when part of the homo- or hetero- oligomeric complexes, are essential for proper binding of the client proteins. The dynamism of the apical domains was assessed by root-mean-square-fluctuation (RMSF) of CCT5 variants and wild type under the three conditions: nucleotide-free, ATP-bound, and ADP-bound. The amino acids of the apical and intermediate domains were the most fluctuating. Notably, the apical and intermediate domains of the Leu224Val variant fluctuated mostly in the absence of nucleotides and in the presence of ADP (Figure 2A,C), whereas the apical and intermediate domains of the His147Arg variant showed the most dynamism in the ATP-bound form (Figure 2B). The equatorial amino acids of the two mutants retained the same wild-type stability under the three conditions tested.

### 2.3. The Number of Hydrogen Bonds Differ in the Two Mutant and the Wild-Type Molecules

Alteration of the hydrogen-bond network of molecular chaperones may lead to changes in domain communication, impair peptide binding, and alter ATP hydrolysis capacity. Therefore, we analyzed the predicted number of hydrogen bonds in CCT5 mutants. The two variants showed different amounts of hydrogen bonds in all three conditions, and both differed from the wild type (Table 1).

The His147Arg variant showed the highest number of hydrogen bonds in all conditions. The number of hydrogen bonds in the wild type was highest in the ADP-bound conformation (Table 1, row 1). In the Leu224Val mutant, hydrogen bonds also increased in the ATP-bound and ADP-bound conditions (Table 1, row 2). There were no differences in the number of hydrophobic residues between wild type and mutants under the three conditions tested, probably due to the conservative nature of the substituted amino acids. We looked for hydrogen bonds around positions 147 and 224 of the ATP-bound molecules, but no hydrogen bond was found for Leu224 (wild type) or Val224 (mutant). Instead, Arg147, in the His147Arg variant, showed more hydrogen bonds (Table 2), with a ratio of 2:4, than the His147 in the wild-type molecule (Table 3). No clear differences were apparent between the wild type and the His147Arg variant in the length of the hydrogen bonds around position 147 (Table 2 and Table 3).

### 2.4. Electrostatic Potential of ATP-Bound CCT5 Variants

Adaptive Poisson–Boltzmann Solver (APBS) software was used to reveal the electrostatic potential of the ATP-bound wild-type CCT5 and the two mutants His147Arg and Leu224Val. The wild-type molecule showed a homogeneous distribution of charges on its surface (Figure 3A); areas with neutral (white) and negative (red) charges were present all over the molecular surface (see Appendix A), while a few positive areas (blue) can be seen in the deep, hidden areas of the molecule. The two CCT5 variants displayed a different distribution of charges on their surfaces: (1) compared to the wild-type subunit, many more positive and negative areas were interspersed on the surface of the variants (Figure 3B,C). The apical areas of the variants presented positive charges not detected in the corresponding apical region of the wild-type molecule; (2) comparing the two variants to each other, the Leu224Val mutant subunit (Figure 3B) had a more negative “dorsal area” (constituting the outer surface of the subunit within the hexadecameric complex), similar to that of the wild-type subunit, unlike the other mutant (His147Arg) in which the same area appeared more neutral (Figure 3C).

## 3. Discussion

The data reported here show that two different missense mutations, His147Arg and Leu224Val, occurring in the equatorial and intermediate domains of CCT5, respectively, can alter the molecular anatomy and physicochemical properties of the subunit. Molecular dynamic simulations revealed the bending of the apical domain toward the equatorial domain of the His147Arg mutant, particularly under nucleotide-free and ATP-bound conditions. On the other hand, the Leu224Val mutation, occurring in the intermediate domain, in the absence of nucleotide, stretches the apical domain, compared to the wild-type subunit, moving the apical domain of the mutant molecule away from its equatorial domain. In the presence of ATP, the apical domain of the Leu224Val mutant returns closer to its equatorial domain than the apical domain of the wild-type subunit, but not as close as the apical domain of the CCT5 His147Arg variant. In the presence of ADP, the molecular models of both variants are similar but differ from that of the wild type. The models show a marked conformational change of the apical domain in both variants, particularly under ATP-bound and nucleotide-free conditions.

The RMSF of both CCT5 mutant subunits showed flexibility/fluctuation of the subunit domains, in agreement with data in the literature [27]. However, we found, in the open state of the Leu224Val mutant (nucleotide-free, Figure 2A and Appendix A), excessive fluctuation of its apical and intermediate domains. Instead, a higher fluctuation of the apical and intermediate domains of the His147Arg variant subunit was observed when bound to ATP. In addition, the apical and intermediate domains of both variants moved differently than the wild-type subunit in the presence of ADP.

The apical domains are the most heterogeneous regions of the CCT subunits and have the role of recognizing, interacting, and capturing the substrates [28]. The apical domain begins the folding of client proteins and its helical protrusion at the end acts as a built-in lid that closes the inner chamber and isolates the substrate [28]. CryoEM studies showed that the apical domains of the CCT subunits acquire a conformation that makes them receptive for the interactions with substrates, but that the apical domains have limited movement [29]. The important role of the apical domain becomes apparent when the CCT is deprived of its apical domains: it is unable to fold substrates and the presence of ATP induces the closure of the CCT chamber; ATP binding to the equatorial domain is not sufficient to close the chamber and isolate the protein-folding process. This suggests that apical domains may perform an allosteric modulation of the chaperonin [30,31].

The different molecular conformations observed between the two variants and the wild-type CCT5 suggest an unusual and instable “starting receptive structure” of the mutants and an abnormal displacements/dynamism of the variants’ domains, especially of the apical domain.

The Leu224Val mutation occurs in the β4-sheet at a point that may be considered the intersection between the intermediate and apical domains. The intermediate domain works as a molecular hinge that bidirectionally transmits and integrates the various conformational changes at the apical and the equatorial domains [32]. To date, there are no data on the ability of the Leu224Val CCT5 mutant to form homo- or hetero-oligomeric complexes, but we know that the His147Arg mutant can form homo-oligomeric complexes [21]. Therefore, it can be assumed that the Leu224Val mutation impairs the functions of the monomers and/or polymeric complex assembly, e.g., by inhibiting the rotation of the equatorial domain during ATP hydrolysis and protein folding. The His147Arg mutation, which occurs in the equatorial domain, could allosterically alter the apical domain, impairing the ability of the CCT5 subunit and, presumably, also that of the CCT5 homo-oligomeric complex, to efficiently recognize and fold client proteins [21]; the mutation may also impede the formation of functional hetero-oligomeric complexes.

We detected other interesting differences between the variants and the wild-type CCT5. Leu and Val are two non-polar hydrophobic amino acids with similar molecular weight, 131.175 g/mol and 117.148 g/mol, respectively. On the other hand, His and Arg are two polar amino acids with a molecular weight of 155.157 g/mol for His, which can, because of its aromatic ring, switch from a positively charged to neutral form, and 174.204 g/mol for Arg, which is always positively charged. Despite the similarities between the substituted amino acids, Leu with Val, and His with Arg, we detected diverse amounts of intramolecular hydrogen bonds in the CCT5 variants (Table 1), suggesting that the mutant amino acids could cause alterations in the steric hindrance of the surrounding amino acids and charge distribution. Hydrogen bonds have a great impact on self-assembly, substrate binding and folding by controlling key molecular features, such as conformation, stability, dynamics, display of surface functional groups, and responsiveness to various stimuli [33]. The three-dimensional structure, known as the native state, reached by proteins to perform their biological activity mainly depends on hydrophobic interactions and intramolecular hydrogen bonding. These two features, and their proper balance, are critical for the correct folding of proteins; their alterations have been linked to the formation of pathogenic amyloid structures, such as those occurring in Parkinson’s and Alzheimer’s diseases [34]. We noted no differences in the hydrophobicity of the CCT5 variants compared to the wild-type molecule. However, in addition to the general alteration of hydrogen bonds of the CCT5 variants, we found a double capability of Arg to locally form hydrogen bonds with adjacent amino acids, likely a consequence of the different structure of the Arg side-chain compared with that of His, whereas no alteration of hydrogen bonds was found for Leu224 (wild type) or Val224 (mutant). We suggest, as a working hypothesis, that the predicted morphological changes discussed here are related to the different distributions of hydrogen bonds in CCT5 variants and that, in turn, allosteric and functional alterations may occur [35].

The charge surface of proteins can act as a direct signal to the proteasome and induce the formation of a specific hydrated surface around the protein [36]. Water molecules can form a complex network that dynamically influences the mobility of defined amino acids that are required for the specific functions of the proteins [37]. The water monolayer covering the surface of the protein causes changes in entropy and determines the well-defined final three-dimensional geometry of the molecule [38,39]. We show that the electrostatic potentials created by charged atoms and hydrogen bonds are profoundly altered in the CCT5 mutants (Figure 3). We suggest that, along with the structural alterations previously discussed, the possible biophysical disturbances of mutated CCT5 subunits may contribute to the dysfunction of the monomer and, if formed, that of the CCT complex.

## 4. Materials and Methods

### Molecular Dynamics Simulation, RMSF, and APBS Analyses

The structural properties of the monomeric CCT5 subunits, wild type, and mutant, were extracted from the structure of the crystallized homo-oligomeric complex deposited in the Protein Data Bank with accession code 5UYZ [25].

The structure of the mutant subunit was obtained by changing amino acid 224 leucine with valine, using the software UCSF Chimera, developed by the Resource for Biocomputing, Visualization, and Informatics at the University of California, San Francisco, with support from NIH P41-GM103311 (accessed on 18 December 2020), [40]. Molecular dynamics (MD) simulations were performed for at least 150 ns and in some cases extended to 300 ns, using the GROMACS 2021.4 package [41,42]. The structures shown have been selected by cluster analysis [43] performed by the g cluster tool implemented in the GROMACS package. Interactions were described using an all-atoms CHARMM27 force-field [44,45]. MD simulations for the various molecules and conditions were as described in [16]. After the equilibration phase, the system was run for a total of 150 ns (up to 300 ns for selected systems) for an NVT production run; the trajectory was saved at a frequency of 10 ps to evaluate dynamical and structural properties. The MD simulations were always checked versus the root-mean-square-fluctuation (RMSF) and the energy profile (data not reported). APBS (Adaptive Poisson–Boltzmann Solver) (https://server.poissonboltzmann.org/) was used for the calculation of the potential around our molecule (by solving equations of continuum electrostatics) [46]. Selected images and protein manipulation were performed using VMD [47] and CHIMERA [41]. All the simulations were carried out on a workstation equipped with AMD Threadripper 3960X, 128 GB DDR4 system memory and Nvidia GeForce GTX 2080 TI GPU with 11GB DDR5 memory. Hydrogen bonds were calculated using UCSF Chimera software tools on molecular dynamics simulations of the mutant and wild-type subunits.

## 5. Conclusions

Many rare genetic, including monogenic, disorders are characterized by a heterogeneity of signs and symptoms differing between patients, which makes diagnosis and pinpointing of the etiopathogenic factor difficult. Consequently, patient management and prognostication are extremely challenging. Efforts toward a better understanding of the molecular mechanisms underlying these pathologies are a promising way to make progress in the management of these diseases. Furthermore, since molecular chaperones are expressed by all the cells of the human body, genetic chaperonopathies are expected to occur with a frequency higher than currently believed, but one must be aware that they may be hidden from routine clinical examination. To remedy this weakness of practical medicine, for the benefit of patients and science, chaperonopathies must be included in medical studies curricula.

The data reported here show that two different missense mutations, His147Arg and Leu224Val, occurring in the equatorial and intermediate domains of CCT5, respectively, are predicted to alter the molecular anatomy and physicochemical properties of the subunit in different ways. The replacement of one amino acid with another, even if both are very similar, is predicted to cause disturbances in nearby charges and bonds that, in turn, seem to propagate to distant sites of the CCT5 molecule, e.g., from a molecular domain (e.g., intermediate domain) to another (e.g., apical domain). These are significant predictions, if we consider that: (1) the apical domain has structural, functional, and regulatory roles; (2) the CCT5 subunit is the starting point for the assemblage of the homo- and hetero-oligomeric complexes; and (3) geometrically specific binding conformations are critical to trigger mechanical inputs to initiate the folding cycle. The pathological result can be predicted to be a defective chaperonin unable to fold substrates, which is most likely the pathogenic mechanism underpinning many of the abnormalities observed in patients’ muscles and nerves.

The predictions emerging from the present study should be a valuable guide to the continuation of in vitro and in vivo studies, expanding those reported previously [15,16,18,19,21,24,25,26], to achieve a complete elucidation of the molecular mechanisms causing the tissue abnormalities observed in patients. For example, simulations, cryoelectron microscopy, and crystallography of the hexadecamer containing the mutant CCT5, either Leu224Val or His147Arg, may provide data on how the structure of the chaperoning complex is affected by the presence of a mutant subunit and, thereby, provide clues to the molecular mechanism underpinning its functional abnormality.

## Figures and Tables

**Figure 1 ijms-24-02018-f001:**
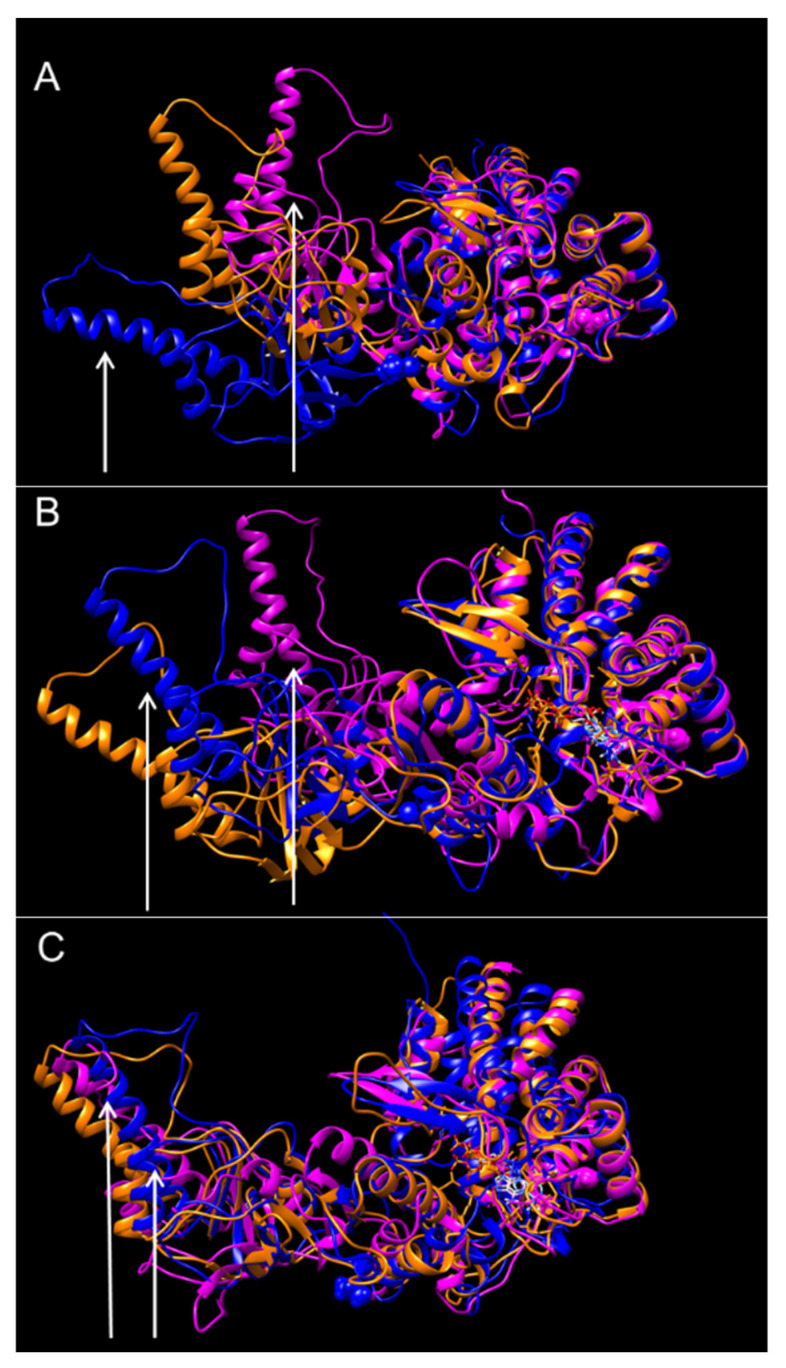
Predicted protein conformations of CCT5 wild type (orange), His147Arg mutant (lilac), and Leu224Val mutant (blue). Shown are the overlays of the most probable conformations obtained from the molecular dynamics simulation of the three proteins. (**A**) nucleotide free; (**B**) ATP-bound; (**C**) ADP-bound. The images show that the mutant proteins have different conformations than the wild type, especially under nucleotide-free and ATP-bound conditions, with the apical domain being the most affected. Arrows indicate the apical domain of the mutant proteins His147Arg (lilac), and Leu224Val (blue).

**Figure 2 ijms-24-02018-f002:**
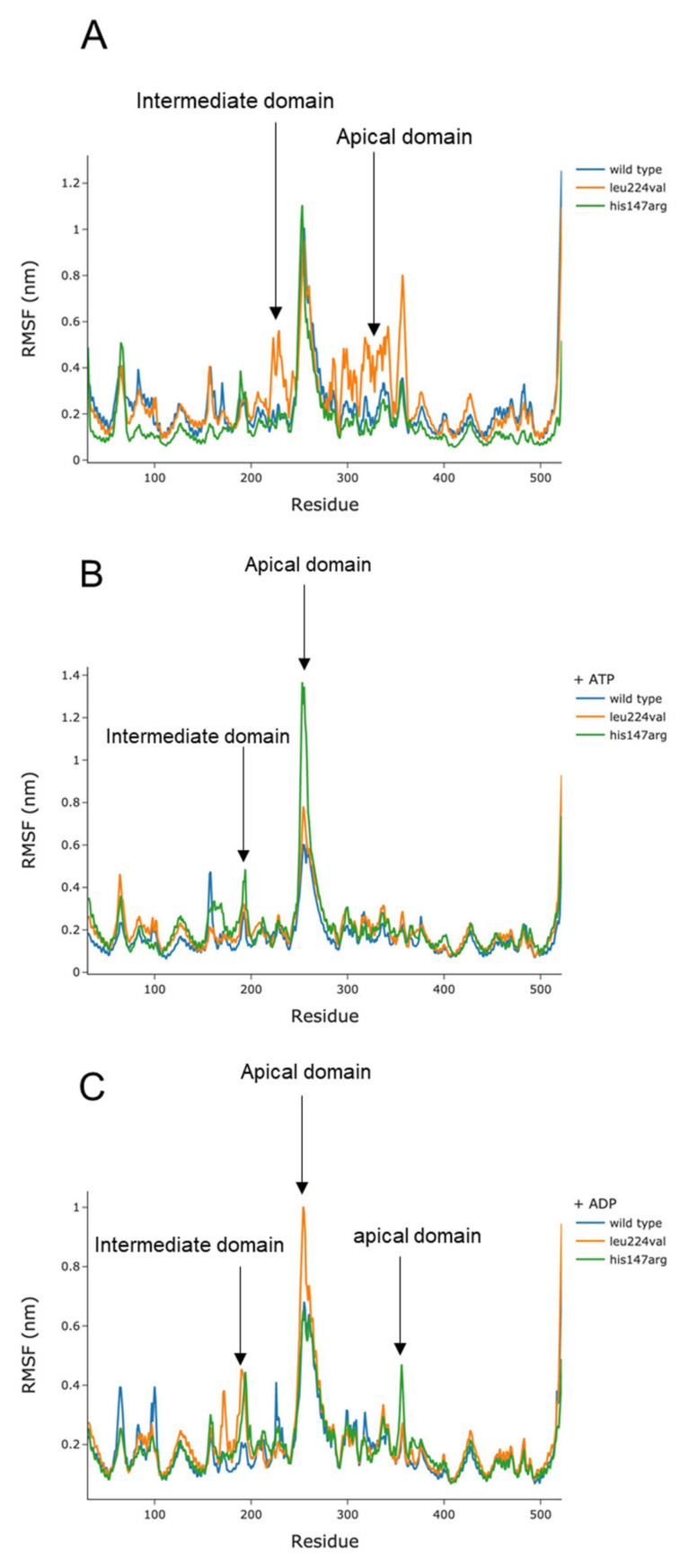
Root-mean-square fluctuation (RMSF) analysis of the amino acids of CCT5 wild-type subunit (blue line), and the His147Arg (green line) and Leu224Val (orange line) mutants. (**A**,**B**,**C**): nucleotide-free, ATP-bound, and ADP-bound conditions, respectively. The images show that the residues corresponding to the apical (amino acids 227 to 378) and intermediate (amino acids 155 to 226 and 381 to 417) domains of both CCT5 variants fluctuate more than wild-type residues under all three conditions. The apical and intermediate domains of the Leu224Val mutant display the greatest dynamism in the absence of nucleotide and in the presence of ADP (**A**,**C**). The apical and intermediate domains of the His147Arg mutant show the highest dynamism in the presence of ATP (**B**). The equatorial domains of the two CCT5 variants, in all three conditions, remain similar to the equatorial domain of the wild-type molecule. See Appendix A.

**Figure 3 ijms-24-02018-f003:**
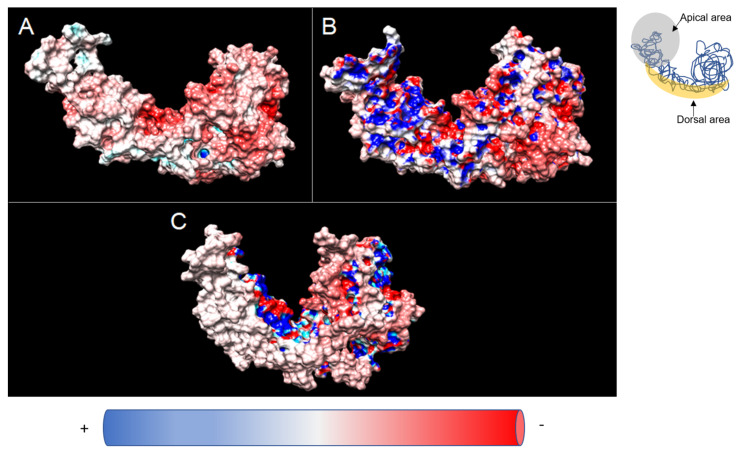
Electrostatic potential of ATP-bound CCT5 subunits. (**A**,**B**,**C**): wild-type, and mutants Leu224Val and His147Arg, respectively. The scale applied ranges from −10 kT/e to +10 kT/e, where k is the Boltzmann constant, T the temperature in K, and “e” the charge. Neutrally charged areas are white, negatively charged areas are red, and positively charged areas are blue. See Appendix A.

**Table 1 ijms-24-02018-t001:** Number of hydrogen bonds in CCT5 wild type and its two mutants under three conditions: nucleotide-free, ATP-bound, and ADP-bound.

CCT5	Nucleotide Free	ATP Bound	ADP Bound
Wild-type Leu224	417	418	426
Mutant Leu224Val	408	423	427
Mutant His147Arg	436	435	435

**Table 2 ijms-24-02018-t002:** Hydrogen bonds of Arg 147 in the His147Arg CCT5 variant ATP bound. Constraints relaxed by 0.4 angstroms and 20 degrees ^1^.

Donor	Acceptor	Hydrogen	D..A dist.	D-H..A dist.
ARG 147 N	VAL 143 O	ARG 147 HN	2.888	2.021
ARG 147 NE	SER 428 O	ARG 147 HE	3.263	2.728
ARG 147 NE	SER 428 OG	ARG 147 HE	2.916	1.960
ILE 151 N	ARG 147 O	ILE 151 HN	3.133	2.190

^1^ Abbreviations: D..A dist., donor-acceptor distance; D-H..A dist., hydrogen of donor-acceptor distance; ARG, arginine; N, nitrogen; VAL, valine; O, oxygen; SER, serine; ILE, isoleucine; H, hydrogen; HN (ARG), hydrogen bonded to nitrogen of arginine backbone; NE (ARG), nitrogen of the guanidino group of the arginine side-chain; HE (ARG), hydrogen bonded to guanidino group of the arginine side-chain; OG (SER), hydroxyl oxygen of the side group of serine; HN (ILE), hydrogen bonded to nitrogen of isoleucine backbone.

**Table 3 ijms-24-02018-t003:** Hydrogen bonds of His147 in the wild-type CCT5 subunit ATP bound. Constraints relaxed by 0.4 angstroms and 20 degrees ^1^.

Donor	Acceptor	Hydrogen	D..A dist.	D-H..A dist.
HIS 147 N	VAL 143 O	HIS 147 HN	2.968	2.063
ILE 151 N	HIS 147 O	ILE 151 HN	2.765	1.776

^1^ Abbreviations: D..A dist., donor-acceptor distance; D-H..A dist., distance between hydrogen of donor and acceptor atom; HIS, histidine; N, nitrogen; VAL, valine; O, oxygen; ILE, isoleucine; HN (HIS), hydrogen bonded to imidazole nitrogen of histidine; HN (ILE), hydrogen bonded to nitrogen of isoleucine backbone.

## Data Availability

The data presented in this study are included in the article/Appendix A. Further inquiries can be directed to the corresponding author.

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
