# Peer review of "Structural and Dynamic Disturbances Revealed by Molecular Dynamics Simulations Predict the Impact on Function of CCT5 Chaperonin Mutations Associated with Rare Severe Distal Neuropathies"

_ijms, 2023, doi:10.3390/ijms24032018_

Round 1

Reviewer 1 Report

Reviewer's comments on manuscript ID ijms-2155864

Authors investigate through bioinformatics and MD simulations, the conformational changes, and the physicochemical properties of the chaperonin CCT5 bearing two mutations, His147Arg and Leu224Val, under three conditions in comparison with the wild type one.

As main result they found that the two mutations alter the CCT5 molecular structure (mainly the apical domain) and the distribution of charges. An accordance with what has been observed, the Authors suppose that the chaperonin may not be able to fold the substrate and therefore lead to tissue abnormalities.

In my opinion, the work presented is very basic, well-known and well-established procedures are used, the results are a bit weak and further studies are needed to confirm the proposed hypotheses. The work may be published in International Journal of Molecular Sciences after the authors address my comments below:

Comments:

·       Please rephrase the caption of Figure 1. What do the arrows indicate?

·       In Figure 3 it would be appropriate to report a color scale with the ranges considered to facilitate reading (similar to the one shown in the video S-V2)

·       The meaning of the sentence at page 9, lines 257-260 is unclear. Correct appropriately the punctuation.

·       It is not clear how the hydrogen bond numbers reported in Tables 1-3 were evaluated. Did the authors consider an average value extracted from molecular dynamics simulations or do they refer to the PDB structure with and without mutations? Please state more clearly in the Methods section and in paragraph 2.3.

·       Section 4 'Materials and Methods' is somewhat lacking in useful information for the purpose of reproducibility of the proposed procedure. Have the ligands present in the crystallographic structure (Mg ion and ADP) been removed or retained? Given the simulation times (ns), considerable structural rearrangements could have occurred if the structures had been simulated in explicit water. Why was the MD simulation conducted without considering hydration? Add the above details and a comment regarding the MD simulation conditions.

·       On page 9 the hydration of the protein surface is only mentioned, but it is not clear to me if they are only considerations reported in the literature or something else.

·       I would suggest revising the Conclusions section and putting more emphasis on the results obtained and the future studies that the Authors intend to undertake.

Minor points:

Please correct at page 2 line 82 "CCT5 monomes"

Author Response

Response to Reviewers

The authors thank the reviewers for their insightful comments and useful suggestions for improving the manuscript.

Reviewer 1

comment #1: Please rephrase the caption of Figure 1. What do the arrows indicate?

Authors’ Reply: We explained what the arrows indicate in the revised manuscript.

comment #2: In Figure 3 it would be appropriate to report a color scale with the ranges considered to facilitate reading (similar to the one shown in the video S-V2)

Authors’ Reply: We added the coloured bar, as requested.

comment #3: The meaning of the sentence at page 9, lines 257-260 is unclear. Correct appropriately the punctuation.

Authors’ Reply: We edited the sentence following this Reviewer’s suggestion.

comment #4: It is not clear how the hydrogen bond numbers reported in Tables 1-3 were evaluated. Did the authors consider an average value extracted from molecular dynamics simulations or do they refer to the PDB structure with and without mutations? Please state more clearly in the Methods section and in paragraph 2.3.

Authors’ Reply: We clarified the issue as requested by this Reviewer,  in Materials and Methods.

comment #5: Section 4 'Materials and Methods' is somewhat lacking in useful information for the purpose of reproducibility of the proposed procedure. Have the ligands present in the crystallographic structure (Mg ion and ADP) been removed or retained? Given the simulation times (ns), considerable structural rearrangements could have occurred if the structures had been simulated in explicit water. Why was the MD simulation conducted without considering hydration? Add the above details and a comment regarding the MD simulation conditions.

Authors’ Reply: The reported structures always refer to the most populated cluster, i.e. the most probable structure found after the analysis done with g-cluster in the region where the rmsd is practically constant (this results in different simulation times between 150ns and 300ns depending on of cases). Water was always present and the details of the simulation are also in our previous cited paper. The water model is the well-known TIP3P.

comment #6: On page 9 the hydration of the protein surface is only mentioned, but it is not clear to me if they are only considerations reported in the literature or something else.

comment #7: I would suggest revising the Conclusions section and putting more emphasis on the results obtained and the future studies that the Authors intend to undertake.

Authors’ Reply: We added in the revised manuscript a short paragraph on future  studies, as suggested by this Reviewer.

Minor points: Please correct at page 2 line 82 "CCT5 monomes"

Authors’ Reply: We corrected it.

Reviewer 2 Report

Authors present a Molecular Dynamics-based analysis of the effect of two mutations on the structural properties of the CCT5 chaperonin. Comparison of structure, mean residues fluctuation and surface electrostatic potential with the wild-type protein reveals noticeable differences also far from the site of mutation. Alterations of the apical domain in particular, which is important for substrate recognition and folding assistance, could explain the pathological effects observed with such mutations.

I think that this work is suitable for publication in International journal of molecular sciences, after the following minor issues have been addressed:

1. No bioinformatics analysis seems to appear in the manuscript, as instead mentioned in the abstract and the main text.

2. Was the His147Arg modeled in the Cpn60 homolog instead of CCT5, as stated in the Introduction? If so, please explain why the CCT5 structure was not employed and report such choice also in the Methods section.

3. The “in silico analyses” term appears quite vague. Maybe it could be substituted with “Molecular Dynamics simulations”.

4. How the “most probable conformations” in Figure 1 were evaluated?

5. Please explicitly report in Figure 1 caption that the arrows point to the apical domains.

6. Could the fact that “no differences in the number of hydrophobic residues between wild type and mutants under the three conditions tested” be simply due to the conservative nature of the mutations?

7. Please report that “Arg147 in the His147Arg variant showed more hydrogen bonds” is a likely consequence of the different structure of Arg side chain (and so more hydrogen bond donors), with respect to His.

8. In section 2.4, please give a possible explanation of why a single mutation could affect the electrostatic potential on all the protein surface.

9. Please better show in Figure 3 the “apical” and “dorsal” areas mentioned in section 2.4.

10. Have the simulation been performed in explicit water solution? If so, please add the employed molecular mechanics water model in Methods section.

11. Since the apical domain appears to be quite exposed and mobile, have other clusters been found beside that shown? How much are they populated with respect to the main cluster?

Author Response

Response to Reviewers

The authors thank the reviewers for their insightful comments and useful suggestions for improving the manuscript.

Reviewer 2

comment #1: No bioinformatics analysis seems to appear in the manuscript, as instead mentioned in the abstract and the main text.

Authors’ Reply: We substituted it with “classical molecular dynamics”.

comment #2: Was the His147Arg modeled in the Cpn60 homolog instead of CCT5, as stated in the Introduction? If so, please explain why the CCT5 structure was not employed and report such choice also in the Methods section.

Authors’ Reply: In the introduction the simulation using the Cpn60 is mentioned in relation to the bibliographic reference n26, as explained. Instead, in the first lines of Materials and Methods, we explain that all the variants have been obtained following simulation with the human model 5UYZ: “The structural properties of the monomeric CCT5 subunits, wild type, and mutant, were extracted from the structure of the crystallized homo-oligomeric complex deposited in the Protein Data Bank with accession code 5UYZ [25].”

comment #3: The “in silico analyses” term appears quite vague. Maybe it could be substituted with “Molecular Dynamics simulations”.

Authors’ Reply: We substituted it with “Molecular dynamics simulations”

comment #4: How the “most probable conformations” in Figure 1 were evaluated?

Authors’ Reply: The reported structures always refer to the most populated cluster, i.e. the most probable structure found after the analysis done with g-cluster in the region where the rmsd is practically constant (this results in different simulation times between 150ns and 300ns depending on the case).

comment #5: Please explicitly report in Figure 1 caption that the arrows point to the apical domains.

Authors’ Reply: We explained in the revised manuscript what the arrows indicate.

comment #6: Could the fact that “no differences in the number of hydrophobic residues between wild type and mutants under the three conditions tested” be simply due to the conservative nature of the mutations?

Authors’ Reply: We mentioned this possibility in the text of the revised manuscript.

comment #7: Please report that “Arg147 in the His147Arg variant showed more hydrogen bonds” is a likely consequence of the different structure of Arg side chain (and so more hydrogen bond donors), with respect to His.

Authors’ Reply: We mentioned this possibility in the text of the revised manuscript.

comment #8: In section 2.4, please give a possible explanation of why a single mutation could affect the electrostatic potential on all the protein surface.

Authors’ Reply: It is explained in lines 295-302 of “Discussion”.

comment #9: Please better show in Figure 3 the “apical” and “dorsal” areas mentioned in section 2.4.

Authors’ Reply: We modified Figure 3 to show the apical and dorsal areas, as requested.

comment #10: Have the simulation been performed in explicit water solution? If so, please add the employed molecular mechanics water model in Methods section.

Authors’ Reply: Water was always present and the details of the simulation are also in our previous cited paper. The water model is the well-known TIP3P.

comment #11: Since the apical domain appears to be quite exposed and mobile, have other clusters been found beside that shown? How much are they populated with respect to the main cluster?

Authors’ Reply: The reported structures always refer to the most populated cluster, i.e. the most probable structure found after the analysis done with g-cluster in the region where the rmsd is practically constant (this results in different simulation times between 150ns and 300ns depending on the case).
